# Image Generation and Recognition for Railway Surface Defect Detection

**DOI:** 10.3390/s23104793

**Published:** 2023-05-16

**Authors:** Yuwei Xia, Sang Wook Han, Hyock Ju Kwon

**Affiliations:** 1Department of Mechanical and Mechatronics Engineering, University of Waterloo, 200 University Avenue West, Waterloo, ON N2L 3G1, Canada; 2Department of Automotive Engineering, Shinhan University, 95, Hoam-ro, Uijeongbu-si 11644, Republic of Korea

**Keywords:** railway defect detection, visual inspection, convolutional neural network, dataset expansion

## Abstract

Railway defects can result in substantial economic and human losses. Among all defects, surface defects are the most common and prominent type, and various optical-based non-destructive testing (NDT) methods have been employed to detect them. In NDT, reliable and accurate interpretation of test data is vital for effective defect detection. Among the many sources of errors, human errors are the most unpredictable and frequent. Artificial intelligence (AI) has the potential to address this challenge; however, the lack of sufficient railway images with diverse types of defects is the major obstacle to training the AI models through supervised learning. To overcome this obstacle, this research proposes the RailGAN model, which enhances the basic CycleGAN model by introducing a pre-sampling stage for railway tracks. Two pre-sampling techniques are tested for the RailGAN model: image-filtration, and U-Net. By applying both techniques to 20 real-time railway images, it is demonstrated that U-Net produces more consistent results in image segmentation across all images and is less affected by the pixel intensity values of the railway track. Comparison of the RailGAN model with U-Net and the original CycleGAN model on real-time railway images reveals that the original CycleGAN model generates defects in the irrelevant background, while the RailGAN model produces synthetic defect patterns exclusively on the railway surface. The artificial images generated by the RailGAN model closely resemble real cracks on railway tracks and are suitable for training neural-network-based defect identification algorithms. The effectiveness of the RailGAN model can be evaluated by training a defect identification algorithm with the generated dataset and applying it to real defect images. The proposed RailGAN model has the potential to improve the accuracy of NDT for railway defects, which can ultimately lead to increased safety and reduced economic losses. The method is currently performed offline, but further study is planned to achieve real-time defect detection in the future.

## 1. Introduction

In 2019, over 5 million passengers traveled by railway in Canada [1], and the total number of passengers transported by rail worldwide was estimated at 29,066 million passenger-km [2]. Along with passenger trains, freight transportation by railway has become an indispensable means of transportation in daily life. Given the high capacity of railway transportation, accidents can result in substantial financial losses, pose a lethal threat to passengers, endanger nearby residents, and cause significant environmental damage. Of all of the serious accidents in railway transportation, derailments are the most common [3], and defective railways are the leading cause of such incidents. A recent report by the Transportation Safety Board of Canada highlights that track-related issues caused 37% of the 959 main-track derailments between 2011 and 2020 [3]. To prevent such incidents, it is crucial to carry out regular and impartial inspections of the tracks to detect defects early, which can significantly reduce the chances of rail accidents due to derailments.

Railway defects are classified into several categories, including transverse defects, longitudinal defects, web defects, base defects, surface defects, and defective weld [4,5]. Among these categories, Surface defects are the most common type of railway defects, which lead to other types of defects such as wear and fatigue. Since surface defects can be detected through visual inspection, computer-vision-based artificial intelligence (AI) methods have become increasingly popular for identifying and classifying these defects. For instance, algorithms such as You Only Look Once (YOLO) and region-based convolutional neural networks (RCNNs) have been developed to detect and segment defects in real time on low-shot walls, steel plates, and even transmission lines [6,7,8]. However, as these algorithms rely on supervised learning, there is a challenge in collecting a sufficient amount of image data—particularly for defective samples. Possible methods for alleviating such data insufficiency include engraving physical defects on real railway samples or creating artificial defects on railway images. However, both methods are inefficient and require extensive human intervention and supervision, and engraving physical defects is particularly costly and time-consuming. In contrast, generating artificial defects on images seems feasible if the process can be automated. There have been attempts to create defective images using network-based image processing techniques, such as the generative adversarial network (GAN) developed by Goodfellow et al. [9].

A GAN model typically comprises two networks: a generator network and a discriminator network [10]. The generator network’s primary function is to generate synthetic images based on normal images, while the discriminator network aims to distinguish defective and normal images. Figure 1 shows the flowchart of a GAN model.

The mapping (X) in a GAN model measures the difference between the non-defective dataset (A) and the defective dataset (B). The generator (G) takes the non-defective dataset (A) as the input and produces a set of defective images ($\hat{A}$) based on the mapping (X). Then, both the generated defective images ($\hat{A}$) and the provided defective dataset (B) are fed to the discriminator (D), which generates a comparison result that indicates the differences between the two datasets. During training, both the generator and the discriminator need to be updated based on the comparison result. The generator is rewarded if it produces images that “fool” the discriminator, while the discriminator is penalized if it misclassifies images. When the loss function reaches a balance, the discriminator cannot distinguish between real and synthetic defects [11,12].

The classic GAN model has evolved into various forms and architectures, such as DCGAN [13,14,15,16], Pix2Pix [17,18,19], CycleGAN [20,21,22], and StyleGAN [23,24,25,26,27,28]. There have been several attempts to generate surface defects using these models. The DCGAN, developed by Radford et al. [13] as a direct extension of the vanilla GAN model, is one such example. It is a type of GAN that uses convolutional layers with the LeakyReLU activation function in the discriminator network and convolutional transpose layers with the ReLU activation function in the generator network. The generator takes a vector of input images and generates an image with an RGB color matrix, while the discriminator uses the information from the RGB images to output a scalar probability. Liu et al. [15] applied DCGAN to generate defective images of micro-precision glass-encapsulated electrical connectors. In their study, both the generator and discriminator had five convolutional layers. With their DCGAN model, Liu et al. successfully generated 1500 defective images from 125 real defective and 75 non-defective samples. Another example of using DCGAN for defective image generation can be found in [16], where an image enhancement algorithm was combined with the DCGAN to generate defects on images of magnetic rings. The image enhancement was conducted with a frequency-based bandpass filter. It was reported that their method can produce images with better detail compared to the original GAN model.

The Pix2Pix model, introduced by Isola et al. [17], takes as its input a tuple consisting of a non-defective image and a corresponding target image with defects. The generator is trained using L1 losses measured by the discriminator until convergence [18]. Mertes et al. [19] used the Pix2Pix model to generate synthetic defects on carbon-fiber surfaces in images, showing that the model can produce realistic-looking synthetic defects and improve pixel-based defect classification, with similar performance to conventional data augmentation using normal images.

The CycleGAN model, proposed by Zhu et al. in [20], has two generators and discriminators. Compared to other GAN models with only forward mapping, the CycleGAN model includes one more generator to learn the inverse mapping. The additional generator allows for the introduction of cycle consistency losses, which ensure that the forward and backward transformations between the two domains are consistent, leading to improved model accuracy. Shuanlong Niu [21] developed the surface defect generation adversarial network (SDGAN) using the core concept of cycle consistency loss to generate images with commutator surface defects. They combined the D2 loss with the original cycle consistency loss and found that the SDGAN model could achieve a lower error rate than the original CycleGAN model in terms of defect generation and classification. Du-Ming Tasi et al. [22] proposed a two-stage CycleGAN model for surface defect generation and identification. In the first stage, they used images with and without defects to generate a synthetic image. In the second stage, they used a binary image containing the shape of the defect, which was produced by the accumulated differences between the defect-free and synthetic defect images, to perform defect segmentation using a U-Net architecture. The model achieved an accuracy of over 90% in annotating and segmenting different patterns of defects on a texture surface.

StyleGAN, introduced by Karras et al. [23], makes several modifications to the image generation process. First, both the generator and the discriminator double the image size in width and height during training. Second, the generator uses adaptive instance normalization to convert the output into a Gaussian distribution with a style vector as a bias. Third, the model adds Gaussian noise to each layer to allow for slight variations in the generated style [23,24,25]. Lastly, StyleGAN includes mixing regularization by using two random latent codes to generate a given percentage of images, allowing the model to localize the style to specific image regions. Situ et al. [26] integrated the StyleGAN model with adaptive discriminator augmentation (ADA) to generate synthetic images of surface defects on water sewers. The model was capable of generating high-resolution images. They also employed the freeze discriminator (Freeze-D) approach in combination with the StyleGAN model and found that this yielded better results than using either StyleGAN or StyleGAN-ADA alone. 

The previously discussed models have shown high accuracy and success rates in generating defect patterns for specific applications. However, they may produce defects in the background when applied to target objects that share similar features with the background. This is because these models lack control over the locations where defects are created. To address this issue, RailGAN was developed in this study to accurately generate artificial defects only on the specific object of interest—namely, railways. RailGAN improves upon existing models by providing better control over the defect locations and mitigating the issue of generating defects in the background.

## 2. Materials and Methods

The RailGAN process is composed of two parts: image segmentation and defect generation. The flowchart of the overall process is shown in Figure 2.

It begins with the image segmentation process to differentiate the background and the railway track surface sections. The defect generation algorithm is then applied to the track surface section. Finally, the original background section is added back to fabricate the defective railway images.

Image segmentation techniques are employed to classify each pixel in the image into two different classes: background and railway track surface. There are several techniques to achieve this; in this project, two approaches were considered: image filtering, and convolutional neural network (CNN).

### 2.1. Image Filtering

For image segmentation, the railway images are first imported as greyscale images to reduce the computation time. Then, two layers of 2D convolutional filters are applied to the images.

The first filter is the Gaussian filter, which blurs the image by performing a downsampling, removing details in the background, and emphasizing key features in the image. Applying the Gaussian filter to the railway images softens the sketchy edges of rocks, sand, or bushes in the background, leaving only the track area as the focus for the next steps. After the Gaussian filtering, a Sobel filter is applied to the image to emphasize the changes between the adjacent pixels. This operation effectively enhances the edges, with a threshold value selected based on the railway edge pixel intensity values. However, light reflection from the uneven surface in the background can also be enhanced. Therefore, an additional step of Houghlines transformation is applied to find straight lines in the image. Houghlines transformation detects the points that can form lines in the image based on the criteria of minimum points and maximum gap width between the points. This function returns coordinates that connect the left and right boundaries of both tracks. These coordinates can also provide information on the orientation of the railway track presented in the image.

Figure 3 illustrates the overall process of incorporating image filters for image segmentation, and the output of each convolutional filter is shown in Figure 4.

In this project, the railway images were initially resized to 256 × 56 to reduce the computation time and power. A Gaussian filter with a 5 × 5 kernel and a standard deviation of 5 in the x direction and 0 in the y direction was applied, followed by a Sobel filter with a 3 × 3 kernel that enhanced the vertical edges by representing the first-order x-derivative. A threshold was applied to generate a binary image, which is shown in Figure 4c. Next, the Houghlines transformation was performed to find lines consisting of at least 200 points with a maximum gap of 1. The parameters were selected empirically to highlight the railway edges, which are shown as green lines in Figure 4d,e. The algorithm used the endpoint coordinates to determine the angle between the found line and the vertical axis, which was used to rotate the image and align the railway vertically. The final step was to separate the railway from the background.

### 2.2. Convolutional Neural Network

U-Net was chosen as the CNN for image segmentation in this project specifically because it has demonstrated excellent performance with fewer sample images compared to other CNN models. U-Net was originally developed for biomedical image segmentation and has since become widely used in various image segmentation tasks [27]. Its unique U-shaped architectureallows for both the extraction of high-level features and the preservation of spatial information, which is crucial for image segmentation tasks. Additionally, U-Net employs skip connections between corresponding layers, which helps to combine information from the encoder and decoder pathways, further improving the segmentation accuracy. Overall, these characteristics make U-Net an excellent choice for image segmentation tasks, especially when the sample size is limited.

U-Net consists of two sections: the contracting path and the expansive path. The contracting path, uses convolutional operations followed by downsampling, similar to other convolutional neural networks. The expansive path, on the right half, uses up-pooling convolution and unpadded convolution to upsample the feature map. The symmetric structure of U-Net allows for skip connections to be made between an encoder layer and a decoder layer, which helps to preserve fine-grained details in the output segmentation map. During training, input images and their corresponding segmentation maps are fed into the network, and the neural network learns to segment the input railway images by adjusting its internal parameters.

The overall structure of the RailGAN model with U-Net is shown in Figure 5.

In this application, each layer of the contracting path used two 3 × 3 unpadded convolutions, followed by a 2 × 2 max pooling. This process was repeated three times to downsample the image, with a ReLU activation function applied at each step. In the expansive path, a 2D transposed convolution was applied to upsample the image, which was then concatenated with the corresponding feature map from the contracting path to retain more accurate predictions. This step was followed by two 3 × 3 unpadded convolutions—also repeated three times—with ReLU as the activation function. At the end of the expansive path, a final 1 × 1 convolution layer was used to map feature vectors to classes. The network was trained with the Adam optimizer using a learning rate of 0.0001, with the loss of the network measured by binary cross-entropy (BCE).

To train the network, two sets of images were required: railway images and their corresponding segmentation map outputs. The segmentation map of each image was manually labeled using online software (https://labelstud.io/, accessed on 10 March 2022). The U-Net dataset contained 120 real-time railway images captured by the drone and their corresponding masked images showing the railway track surface. Samples of the created dataset are presented in Figure 6.

After obtaining an ideal model, the images were processed by using Houghlines transformation to detect the boundaries of the railway and separate the track from the background.

### 2.3. Defect Generation

Unlike other GANs that need to pair the images to perform the translation, CycleGAN enables unpaired translation. Taking the railway defects for example, to use other GANs, we need to train the algorithm with the same railway images with and without the defect pattern on the body. The images need to be paired together, and they are sent to the adversarial network to train one pair after the other, which is hard to achieve in our case. However, CycleGAN has the advantage of enabling unpaired image-to-image translation, so the defect generation algorithm in RailGAN is based on CycleGAN. As introduced in the previous section, the potential for the CycleGAN model to outperform the vanilla GAN model in creating artificial defective images is granted by CycleGAN’s delicate architecture. The detailed process is shown in Figure 7.

The typical GAN model has only the forward mapping X from A (non-defect) to B (defect), while the CycleGAN model introduces one more generator, Y, to learn the inverse mapping from B to A. The two discriminators measure the similarity between the original non-defect dataset and the generated defect images, as well as the original defect dataset with the generated defect images. CycleGAN can further improve its model accuracy by using both forward and inverse mappings and exploiting both cycle consistency losses. After dataset $\hat{A}$ is produced by the first generator, $\hat{\hat{A}}$ is created through the inverse mapping F, which allows one cycle consistency loss to be measured for the similarity between the original sample A and the generated $\hat{\hat{A}}$. Similarly, there is one more loss measured between $\hat{\hat{B}}$ and B. These losses ensure the quality of the generators, providing further benefits to the overall performance of the model.

Table 1 shows all of the parameters used for building the discriminator network.

The discriminator network utilized LeakyReLU as its activation function, with a negative slope coefficient of 0.2. The mean square error (MSE) was employed to measure the discriminator losses, and a discount factor of 0.5 was applied to the loss to slow down the discriminator’s learning rate compared to that of the generators. The Adam optimizer was used for training, with a learning rate of 0.0001.

Table 2 details the generator network configuration.

The “c7s1-64” layer is a 7 × 7 normalized layer that uses instance normalization, and its activation function is ReLU. This layer can also be referred to as an instanceNormReLU layer, with 64 filters and stride of 1. The second and third layers—labeled “d”, followed by a number “k” (abbreviated as “dk”)—are 3 × 3 instanceNormReLU layers with “k” filters and stride of 2. The residual layers (R256) consist of two 3 × 3 convolutional layers with 9 residual blocks concatenated to the output channel-wise. The fourth and fifth layers (abbreviated as “uk”) are 3 × 3 fractional-stride convolutional instanceNormReLU layers with “k” filters and stride of 0.5. It is worth noting that, from the fourth layer, the convolutions applied are transposed to convert the generated image back to its original size (256 × 256 × 3). Finally, in the sixth layer (c7s1-3), the Tanh activation function is used with 3 filters and stride of 1.

### 2.4. Defect Identification

In order to evaluate the quality of the fabricated images, a defect identification algorithm was utilized. YOLO was selected for this purpose due to its superior overall performance in terms of both speed and accuracy compared to other object detection algorithms [28]. Unlike other two-stage object detection methods such as RCNN, Faster RCNN, or FPN, YOLO uses a one-stage detection method that combines the region proposal and prediction processes in a single neural network. YOLO’s architecture is based on GoogLeNet, and its architecture has a total of 24 convolutional layers with 2 fully connected layers at the end [29]. The first 24 convolutional layers are used for image classification by extracting the image features. This architecture is similar to that of GoogLeNet; however, YOLO simplifies GoogLeNet by replacing the idea of the inception module with a combination of a 1 × 1 reduction layer with a 3 × 3 convolutional layer—except for the first layer, which has a 7 × 7 filter [30]. The last two fully connected layers are used for prediction. YOLO-v5 uses the LeakyReLU activation function in the hidden layers and sigmoid for the detection layer.

YOLO splits the input image into a grid of S*S cells, and each cell is responsible for predicting B bounding boxes, where B is a hyperparameter set during training. By using small grids and multiple bounding boxes per grid cell, YOLO can detect objects of different sizes and aspect ratios. Each bounding box is represented by 5 values: the coordinates of the center of the box (x,y), the size of the bounding box (width and height), and the confidence score. The confidence score represents the probability that an object is contained within the bounding box. The first four values allow the bounding box to locate the object, but they must be normalized into the range between 0 and 1 to be fed into the CNN. For x and y, these values only need to be divided by the size of the grid. The width and height of the bounding boxes can be normalized by dividing them by the width and height of the whole image. The YOLO network can also predict the probability of a grid cell containing an object and its class-specific confidence score. The object can be located by multiplying the confidence score by the class probability. Equation (1) represents this relationship, where the class probability is conditioned on the object being present in the cell.
(1)$$\mathrm{Pr}\left(\mathrm{Class}|\mathrm{Object}\right)*\mathrm{Pr}\left(\mathrm{obj}\right)*{\mathrm{IOU}}_{\mathrm{truth}}^{\mathrm{pred}}=\mathrm{Pr}\left(\mathrm{Class}\right)*{\mathrm{IOU}}_{\mathrm{truth}}^{\mathrm{pred}}$$
where IOU is the intersection over union between the predicted bounding box and the truth. There can be different grid cells that predict the same object. By setting the threshold of IOU, YOLO can discard one that is lower than the threshold (these grids are considered irrelevant to the class). However, this does not solve the problem of having multiple bounding boxes that contain the same object and are all above the threshold. This can cause noise during the training. To tackle this issue, YOLO uses NMS to keep the boxes that have the largest confidence scores and remove the ones with lower confidence.

There are several versions of YOLO available, but for this case, YOLO-v5 was chosen because it offers improved performance with a smaller network size and is an order of magnitude faster (45 frames per second) than the previous version. YOLO-v5 achieves these characteristics by introducing three main features: a backbone network, neck network, and head network, which are used to establish the overall network structure. The backbone network utilizes CSP-Darknet53 to extract and gather features from input images, using a strategy that employs residual and dense blocks to address the vanishing gradient problem. Additionally, CSP-Darknet shortens the gradient flow to overcome the redundant gradient problem. The neck network combines these features and passes them on to the prediction step using the path aggregation network and spatial pyramid pooling, which are able to create feature pyramids quickly. Finally, the head network applies the anchor box with a bounded box region of prediction, along with class probability, to produce the final output vector.

YOLOv5 uses two types of activation function: SiLU, also known as Swish, is used for the hidden layers due to its better performance in situations between linear and more complex nonlinear activation functions. Sigmoid functions are used for the output convolutional layers. The binary cross-entropy (BCE) loss function is used to compute the class probability and object score.

## 3. Results

### 3.1. Image Segmentation

The U-Net model was evaluated on 20 railway images and their corresponding masked images found online. The accuracy of the network was measured as the total number of correct pixel values over the total number of pixel values in all images and reached 96.44% after 100 epochs. These results were considered satisfactory for the RailGAN model. Sample test images are shown in Figure 8, where the first row depicts original railway images, and the second row shows the masked images created by U-Net.

To compare the effectiveness of U-Net with traditional image filtering techniques for railway track segmentation, both methods were applied to the same set of real-time images, as shown in Figure 9. The left column shows three original railway images, while the segmented results obtained by image filters and U-Net for those images are shown in the middle and right columns, respectively.

When using image filter techniques for railway track segmentation, manual adjustment of threshold values based on the pixel intensity values of the track was required. In Figure 9, the threshold was set between 130 and 220 for the first row and between 170 and 220 for the second row. In the third row, an inverse binary thresholding between 160 and 250 was necessary due to the higher pixel value of the railway track compared to the background. However, miscategorization of some pixels occurred in all three images, and the image processing technique segmented part of the background as well as the railway track region, and vice versa. In contrast, the U-Net approach consistently produced accurate results for all images, regardless of the pixel intensity values of the track surfaces. Therefore, U-Net was selected as the image segmentation solution for the RailGAN model.

### 3.2. RailGAN Implementation

To address the limited dataset of 15 defective railway surface images, various image manipulation techniques were utilized, such as image rotation, resizing, cropping, and contract adjustment, which expanded the input data to 100 images, along with 100 non-defect images, for RailGAN training. The model was saved after every 200 epochs, and its performance was monitored. While the network improved over time, it became unstable after 1800 epochs, producing images with significant color tone changes. Nonetheless, the best performing model from the validation process was the one trained with 1800 epochs. For further evaluation, the model was applied to 20 images taken from real-time drone video, depicting railway tracks from various angles and in various lighting conditions.

To confirm the effectiveness of RailGAN in generating railway track defects compared to the original CycleGAN model, a comparison test between the two models was conducted. First, the CycleGAN model was retrained using the same 200-image dataset (100 defect and 100 non-defect images), but this time without the image segmentation by U-Net. Then, both the RailGAN and CycleGAN models were applied to the same test images, and the results were compared. Figure 10b presents the outputs of the CycleGAN model, with the defects generated not only on the track but also on the background, whereas the RailGAN model generated defects only on the track surface, as shown in Figure 10c. This indicates that the RailGAN model is more effective than the CycleGAN model in generating realistic railway track defects only in the relevant areas.

### 3.3. Defect Identification Algorithm

The RailGAN algorithm was used to generate a dataset of 100 defective images with various crack patterns and lighting conditions. The defects in these fabricated images were then labeled using Roboflow, and the dataset was split into training and validation sets at a 9:1 ratio. Figure 11 shows some examples of the labeled dataset.

These fabricated images, coupled with the coordinates of the corresponding label bounding box in each image, were sent to the YOLO-v5 network by Roboflow for training. The training process was originally set with 2000 epochs at a learning rate of 0.0001, and Roboflow automatically saved the model with the best performance. For YOLO, the performance of the model was evaluated by precision, recall, and mean average precision with a threshold over 0.5 (mAP50); these values are visually presented in Figure 12. During the training process, the loss decreased, while the precision, recall, and mAP50 increased. However, a sign of overfitting was observed, as the precision, recall, and mAP50 slightly decreased towards the end of the training. The model reached the best overall performance after 1348 epochs, which was automatically selected by Roboflow, with 95.8% precision, 83.6% recall, and 91.1% mAP50. This model was saved for testing and validation.

The model was then applied to original defect images to test its effectiveness on real defect images. The real defect images were first resized to 416 × 416. Among the 10 real defect images with cracks found online, the trained YOLO model was able to correctly identify nine defects, with a total speed of 0.4 ms pre-processing time, 7.6 ms inference, and 14.6 ms NMS per image, with the NVIDIA GeForce RTX 3060 graphics card. Examples of the test results are shown in Figure 13.

## 4. Discussion

The RailGAN model proposed in this study showed promising results in generating artificial defect images of railways. When both the RailGAN and CycleGAN models were applied to the same test images, the RailGAN model performed significantly better than the CycleGAN model, as shown in Figure 10. Figure 10b shows that the CycleGAN model has no control over where the defects are generated. In contrast, Figure 10c demonstrates that the RailGAN model generates defects only on the track surface. Moreover, all of the defect images generated by the RailGAN resemble real cracks on the railway surface, indicating that these images can be effectively utilized for the defect identification and classification algorithm in the next stage.

To evaluate the performance of RailGAN, the YOLO network was trained with fabricated images by RailGAN and applied to detect real railway defects. The results were promising, with high precision, recall, and mAP50. However, it should be noted that this evaluation was conducted on a relatively small dataset. Further testing on a larger and more diverse set of real-world images will be necessary to fully assess the RailGAN model’s performance.

## 5. Conclusions and Future Work

This paper proposed the RailGAN model to generate artificial defect images of railways, consisting of track–background separation and defect generation. The model first separated the railway track from the background using two different methods. The first method was based on image filters such as the Gaussian filter and Sobel filter to extract the railway track surface, while the second method adopted U-Net. The network was trained using 120 railway images, along with their labeled masked images. Comparison of the results from the two methods indicated that the U-Net model achieved more consistent results, while the image filter method was highly dependent on the threshold values. Therefore, U-Net was chosen as the image segmentation method for the RailGAN model. The extracted railway track surface was then used for the defect generation, with the defect generation method selected based on CycleGAN, followed by splicing the background section back to complete the entire process.

When the RailGAN model was applied to live footage taken by a drone, it was evident that the model performed significantly better than the original CycleGAN model. RailGAN could successfully generate defects only on the railway surface, while the original CycleGAN model tended to generate defects in irrelevant areas.

As it is challenging to assess the quality of the images using statistics, the quality of the fabricated images was evaluated by training a defect identification algorithm with the images generated by the RailGAN model. The YOLO network was used to detect defects in real railway images offline, and the results showed high precision, recall, and mAP50. Future work will involve applying the defect identification algorithm to a larger and more diverse dataset, as well as further tuning the network for improved performance. The ultimate goal is to achieve real-time defect detection using the proposed method on the drone. Overall, the proposed RailGAN model has the potential to enhance railway inspection and maintenance by facilitating automated defect detection and classification.

## Figures and Tables

**Figure 1 sensors-23-04793-f001:**
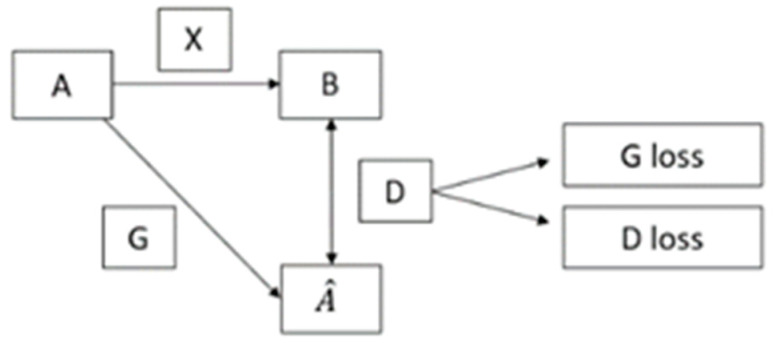
Flowchart of the GAN model.

**Figure 2 sensors-23-04793-f002:**
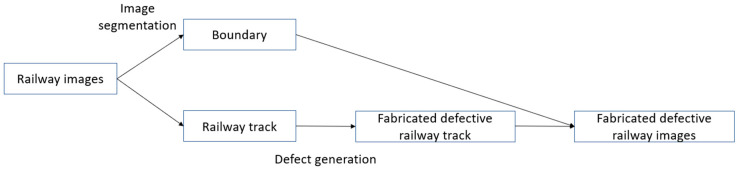
Process of RailGAN.

**Figure 3 sensors-23-04793-f003:**
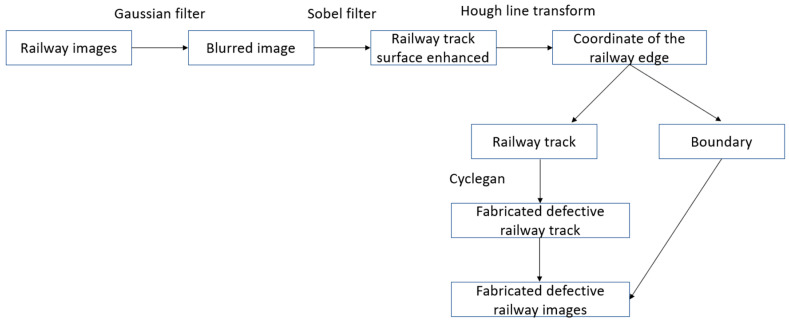
RailGAN process with image filter.

**Figure 4 sensors-23-04793-f004:**
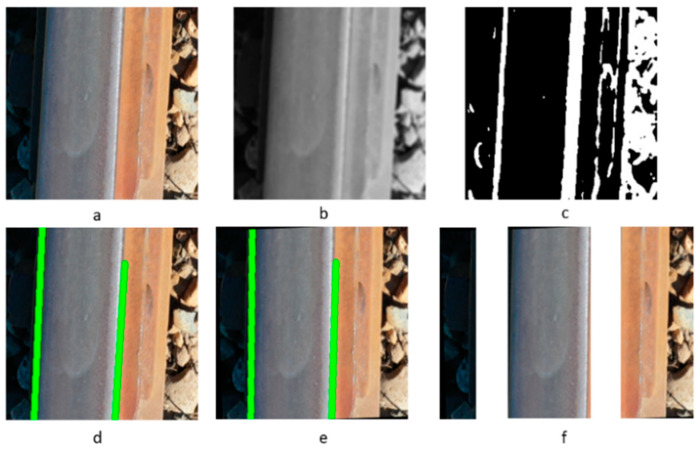
(**a**) Live image, (**b**) greyscale image from Gaussian filter, (**c**) binary image from Sobel filter, (**d**) lines detected by Hough transformation, (**e**) straightened image after rotation, and (**f**) Image segmentation after track and background identification.

**Figure 5 sensors-23-04793-f005:**
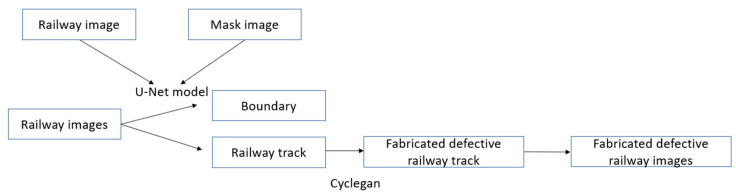
RailGAN with U-Net.

**Figure 6 sensors-23-04793-f006:**
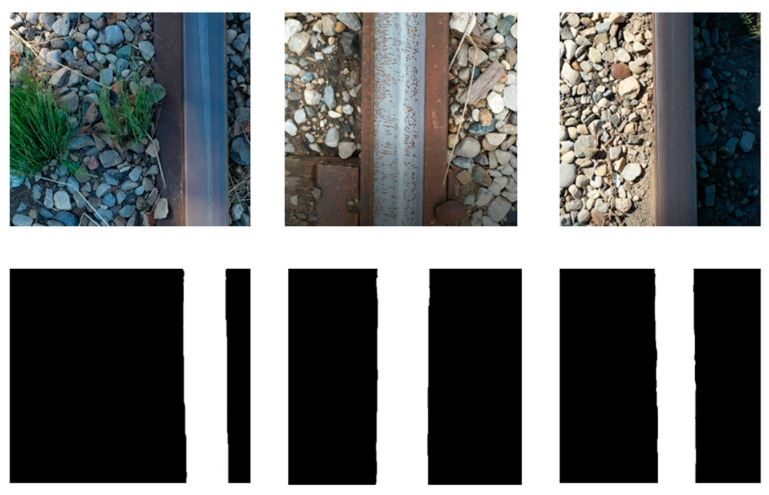
Real-time railway images (**top**) and masked images (**bottom**).

**Figure 7 sensors-23-04793-f007:**
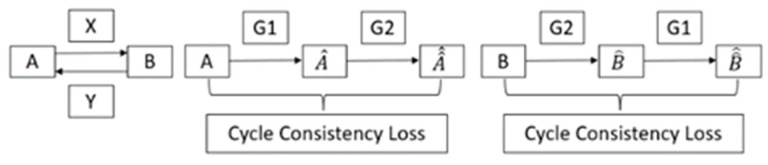
CycleGAN model architecture.

**Figure 8 sensors-23-04793-f008:**
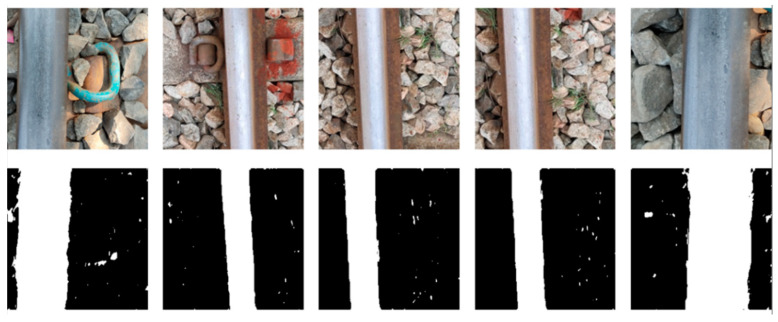
Results of U-Net after 100 epochs.

**Figure 9 sensors-23-04793-f009:**
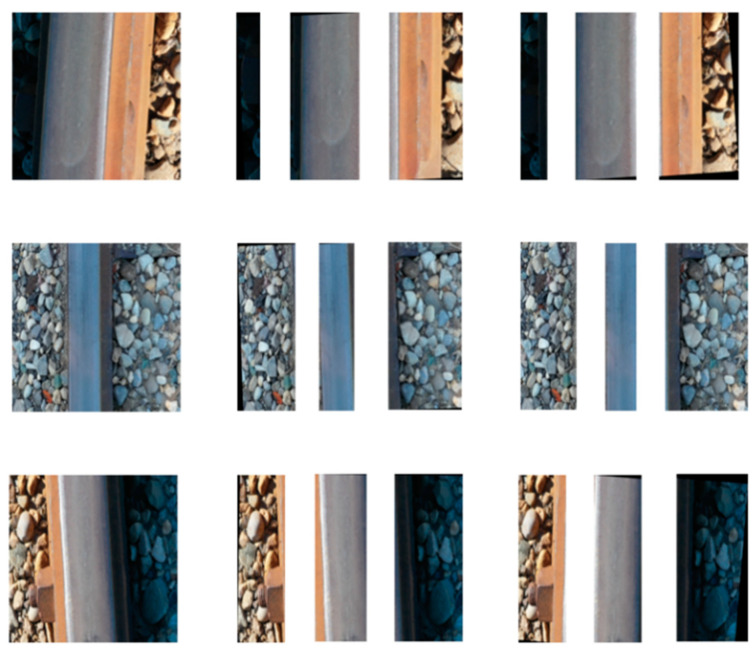
(**Left**) column: original images; (**center**) column: image segmentation by image filters; (**right**) column: image segmentation by U-Net.

**Figure 10 sensors-23-04793-f010:**
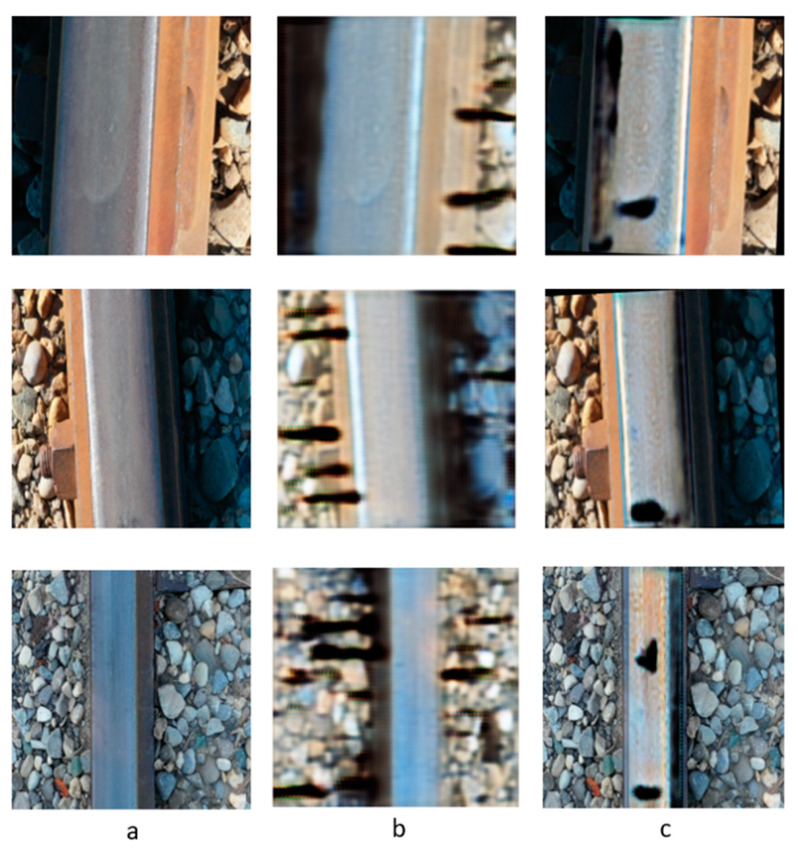
Three example images with different lighting conditions and angles: (**a**) live image of railway track; (**b**) results from the original CycleGAN model; (**c**) results from the RailGAN model.

**Figure 11 sensors-23-04793-f011:**
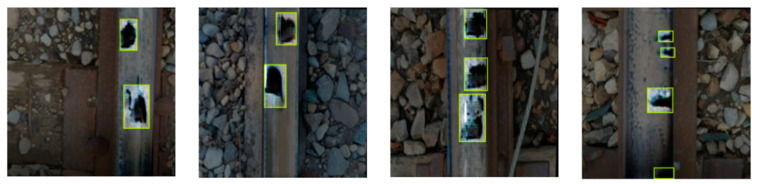
Examples of labeled defect images.

**Figure 12 sensors-23-04793-f012:**
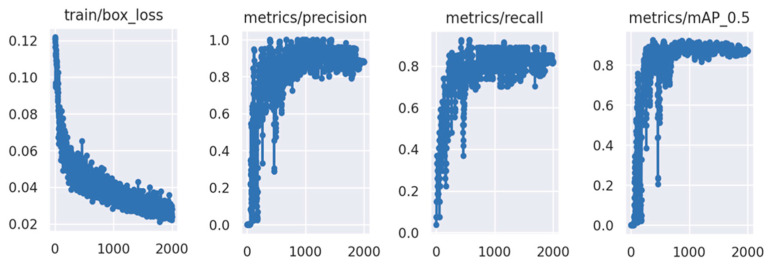
Loss, precision, recall, and mAP50 during training.

**Figure 13 sensors-23-04793-f013:**
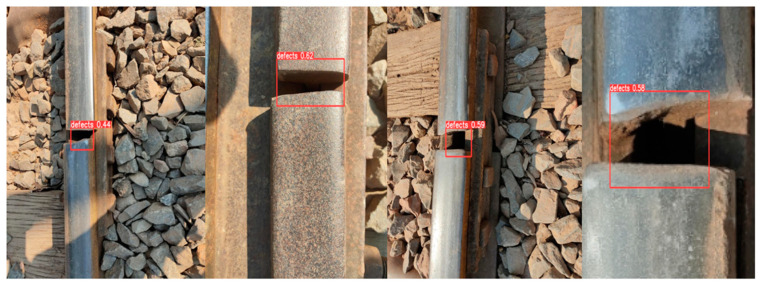
Examples of crack defects identified by the YOLO algorithm.

**Table 1 sensors-23-04793-t001:** Discriminator network parameters.

	Input Size	Output Size
Input Layer	N/A	256, 256, 3
First Layer (c7s1-64)	256, 256, 3	128, 128, 64
Second Layer (c128)	128, 128, 64	64, 64, 128
Third Layer (c256)	64, 64, 128	32, 32, 256
Fourth Layer (c512)	32, 32, 256	16, 16, 512
Second-Last Output Layer	16, 16, 512	16, 16, 1
Output Layer	16, 16, 1	N/A

**Table 2 sensors-23-04793-t002:** Generator network parameters.

	Input Size	Output Size
Input Layer	N/A	256, 256, 3
First Layer (c7s1-64)	256, 256, 3	256, 256, 64
Second Layer (d128)	256, 256, 64	128, 128, 128
Third Layer (d256)	128, 128, 128	64, 64, 256
Residual Layer (R256)		
Fourth Layer (u128)	64, 64, 256	128, 128, 128
Fifth Layer (u64)	128, 128, 128	256, 256, 64
Sixth Layer(c7s1-3)	256, 256, 64	256, 256, 3
Output Layer	256, 256, 3	N/A

## Data Availability

The data presented in this study are openly available in https://github.com/DvdXxxiia/RainGAN (accessed on 10 March 2022).

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
