# Peer review of "Image Generation and Recognition for Railway Surface Defect Detection"

_sensors, 2023, doi:10.3390/s23104793_

Round 1

Reviewer 1 Report

the proposed framework has some novelty in creating a defect rail image for the training of the deep learning model. The main deficiency of the paper is that it is for testing it on real images, they test it on 10 images which is a very small dataset in the domain of image processing. They should test it on a larger and more diverse set of real-world images necessary to assess the model's performance fully.

Author Response

The proposed framework has some novelty in creating a defect rail image for the training of the deep learning model. The main deficiency of the paper is that it is for testing it on real images, they test it on 10 images which is a very small dataset in the domain of image processing. They should test it on a larger and more diverse set of real-world images necessary to assess the model's performance fully.

R) To address the concern regarding the limited number of real-world images used for testing, we agree that a larger and more diverse dataset would be necessary to fully assess the performance of the model. However, it is important to note that in the real world, defective railway images are rare and difficult to obtain. The primary purpose of this study was to develop the RailGAN model to expand the scarce dataset of defect railway images. Some of the real-world images we found were used to train the RailGAN model, and we did not want to use the same dataset again for evaluating the quality of the images. Nonetheless, we acknowledge the importance of evaluating the model's performance on a larger dataset, and we plan to do so in future work.

Reviewer 2 Report

In this article, the RailGAN model generates artificial defect images of railways, using a track-background separation and defect generation process. The model uses image filters or U-Net for track-background separation, and CycleGAN for defect generation. RailGAN performs better than the original CycleGAN model and is used to train a YOLO network for defect detection. The proposed RailGAN model could improve railway inspection and maintenance through automated defect detection and classification. Future work includes applying the algorithm to a larger dataset and improving network performance.

It is a well structured article with further work.

Minor typos and grammarly corrections are needed but could be readable.

Author Response

In this article, the RailGAN model generates artificial defect images of railways, using a track-background separation and defect generation process. The model uses image filters or U-Net for track-background separation, and CycleGAN for defect generation. RailGAN performs better than the original CycleGAN model and is used to train a YOLO network for defect detection. The proposed RailGAN model could improve railway inspection and maintenance through automated defect detection and classification. Future work includes applying the algorithm to a larger dataset and improving network performance.

It is a well-structured article with further work.

Minor typos and grammarly corrections are needed but could be readable.

R) We appreciate reviewer’s feedback. Thank you for pointing out the minor typos and grammar errors. We have gone through the manuscript again and made the necessary corrections to improve readability. Thank you for your feedback, and we hope the revised version meets your expectations.

Reviewer 3 Report

The authors propose the RailGAN model, which enhances the basic CycleGAN model by introducing a pre-sampling stage for railway tracks. Positive images are segmented and the artificial defects are generated only in the track area detected. The idea is interesting, but some points must be better explained in order to justify the importance of the proposed solution. The work is interesting and the paper is well writen, with only a few minor errors found and listed in sequence.

At first, a conceptual question. What are the so called "real-time railway images" mentioned by the authors? Usually, real-time is applied to something that happens at the time of the capture. At first I thought the detection of defects was happening in real time, as soon as the image frames were detected, but it does not seem to be the case. The images were captured using a drone and they are processed later. How to the authors imagine the proposed work being applied in a real case scenario?

The authors do not provide information regarding computational performance, so it is not possible to know if the proposed algorithm is real time. YOLOv5 is known for its real time performance, but in the case of this paper, please provide more information about it.

I believe the weakest point of this work is that it used CycleGAN as base to develop RailGAN and in sequence compares the performence of the proposed approach only to CycleGAN. It would be nice to compare the proposed approach to the state of the art as well, highlighting the advantages and limitations of the proposed approach compared to the others found.

I believe some important references are missing in the paper, such as:

- Deep Learning for Railroad Inspection (https://www.pavemetrics.com/wp-content/uploads/2020/10/Deep-Learning-for-Railroad-Inspection-May-2018-003.pdf)

- Rail track condition monitoring: a review on deep learning approaches (https://oaepublishstorage.blob.core.windows.net/7cb2c33c-13bc-40ef-8ac0-4d653a4180fd/4477.pdf)

- Deep convolutional neural networks for detection of rail surface defects (https://www.dcsc.tudelft.nl/~bdeschutter/pub/rep/16_002.pdf)

- The Recent Applications of Machine Learning in Rail Track Maintenance: A Survey (https://ris.utwente.nl/ws/portalfiles/portal/167171939/ChenariyanNakhaee2019recent.pdf)

- Developing Machine Learning-Based Models for Railway Inspection (https://www.proquest.com/openview/c26a284ae0f27d5b1df012b466ce9e03/1?pq-origsite=gscholar&cbl=2032433)

- Rail-5k: a Real-World Dataset for Rail Surface Defects Detection (https://arxiv.org/pdf/2106.14366.pdf)

More general comments and minor errors are listed as follows.

"mapping, The" -> "mapping. The"

"Figure 9 [30]. its" -> "Figure 9 [30]. Its"

"its superior overall performance in terms of both speed and accuracy compared to other object detection algorithms." -> this sentence seems incomplete

"these grids is considered" -> "these grids are considered"

"the boxes that has" -> "the boxes that have"

"While the network improved over time, becoming unstable after 1800 epochs, producing images with significant color tone changes." -> this sentence seems incomplete

Author Response

1) The authors propose the RailGAN model, which enhances the basic CycleGAN model by introducing a pre-sampling stage for railway tracks. Positive images are segmented and the artificial defects are generated only in the track area detected. The idea is interesting, but some points must be better explained in order to justify the importance of the proposed solution. The work is interesting and the paper is well written, with only a few minor errors found and listed in sequence.

At first, a conceptual question. What are the so called "real-time railway images" mentioned by the authors? Usually, real-time is applied to something that happens at the time of the capture.

R) We appreciate reviewer’s feedback. The “real-time railway images” mean the images that are taking by the drone. Because this project wants to tackle the problem of real-time defect detection, so for the next step, the defect detection algorithm will be trained with the fabricated images that resembles those "real-time" images.

2) At first I thought the detection of defects was happening in real time, as soon as the image frames were detected, but it does not seem to be the case. The images were captured using a drone and they are processed later. How to the authors imagine the proposed work being applied in a real case scenario?

The authors do not provide information regarding computational performance, so it is not possible to know if the proposed algorithm is real time. YOLOv5 is known for its real time performance, but in the case of this paper, please provide more information about it.

I believe the weakest point of this work is that it used CycleGAN as base to develop RailGAN and in sequence compares the performence of the proposed approach only to CycleGAN. It would be nice to compare the proposed approach to the state of the art as well, highlighting the advantages and limitations of the proposed approach compared to the others found. 

I believe some important references are missing in the paper, such as:

- Deep Learning for Railroad Inspection (https://www.pavemetrics.com/wp-content/uploads/2020/10/Deep-Learning-for-Railroad-Inspection-May-2018-003.pdf)

- Rail track condition monitoring: a review on deep learning approaches (https://oaepublishstorage.blob.core.windows.net/7cb2c33c-13bc-40ef-8ac0-4d653a4180fd/4477.pdf)

- Deep convolutional neural networks for detection of rail surface defects https://www.dcsc.tudelft.nl/~bdeschutter/pub/rep/16_002.pdf)

- The Recent Applications of Machine Learning in Rail Track Maintenance: A Survey (https://ris.utwente.nl/ws/portalfiles/portal/167171939/ChenariyanNakhaee2019recent.pdf)

- Developing Machine Learning-Based Models for Railway Inspection (https://www.proquest.com/openview/c26a284ae0f27d5b1df012b466ce9e03/1?pq-origsite=gscholar&cbl=2032433)

- Rail-5k: a Real-World Dataset for Rail Surface Defects Detection (https://arxiv.org/pdf/2106.14366.pdf)

R) We appreciate reviewer's feedback. Regarding the real-time detection of defects, we apologize for any confusion. The images captured by the drone are processed offline in this article, and the proposed algorithm is not designed for real-time defect detection in the current stage, but it is considered in our future work. We have clarified this in the manuscript in abstract and conclusion.

Regarding the computational performance, we have added more information on the processing time and hardware used in our experiments to provide a better understanding of the computational performance in section 3.3.

Regarding the comparison of our proposed approach to other state-of-the-art methods, we have added a justification in section 2.3 on why we chose CycleGAN as the base model for our approach. We have also included additional references that were suggested by the reviewer. We appreciate your feedback and believe that these revisions have improved the quality of our manuscript.

3) More general comments and minor errors are listed as follows. 

"mapping, The" -> "mapping. The"

"Figure 9 [30]. its" -> "Figure 9 [30]. Its"

"its superior overall performance in terms of both speed and accuracy compared to other object detection algorithms." -> this sentence seems incomplete

"these grids is considered" -> "these grids are considered"

"the boxes that has" -> "the boxes that have"

"While the network improved over time, becoming unstable after 1800 epochs, producing images with significant color tone changes." -> this sentence seems incomplete

R) We appreciate the reviewer's detailed comments and have carefully gone through the manuscript to address the pointed-out errors. We have corrected the typos and grammar mistakes, including those mentioned in the review.

Round 2

Reviewer 3 Report

Thank you for the modifications on the text. I'm satisfied with them. Please address the minor issues as follows. After that, I believe the paper can be accepted for publication.

Please improve the quality of Figure 2.

Please improve the quality of Figure 3.

Please improve the quality of Figure 5.

Please improve the quality of Figure 6.

" the other., which " -> " the other, which "

"The reak defect" -> "The real defect"

Author Response

We appreciate reviewer's feedback. We have fixed the pointed-out grammar issues and improved the image quality. Thank you again for kind and detailed feedback.